# Efficient Nonmyopic Bayesian Optimization via One-Shot Multi-Step Trees

**Shali Jiang**[*]
Facebook
shalijiang@fb.com

**Daniel R. Jiang**[*]
Facebook
drjiang@fb.com

**Maximilian Balandat**[*]
Facebook
balandat@fb.com

**Brian Karrer**
Facebook
briankarrer@fb.com

**Jacob R. Gardner**
University of Pennsylvania
jacobrg@cis.upenn.edu

**Roman Garnett**
Washington University in St. Louis
garnett@wustl.edu

## Abstract

Bayesian optimization is a sequential decision making framework for optimizing expensive-to-evaluate black-box functions. Computing a full lookahead policy amounts to solving a highly intractable stochastic dynamic program. Myopic approaches, such as expected improvement, are often adopted in practice, but they ignore the long-term impact of the immediate decision. Existing nonmyopic approaches are mostly heuristic and/or computationally expensive. In this paper, we provide the first efficient implementation of general multi-step lookahead Bayesian optimization, formulated as a sequence of nested optimization problems within a multi-step scenario tree. Instead of solving these problems in a nested way, we equivalently optimize all decision variables in the full tree jointly, in a "one-shot" fashion. Combining this with an efficient method for implementing multi-step Gaussian process "fantasization," we demonstrate that multi-step expected improvement is computationally tractable and exhibits performance superior to existing methods on a wide range of benchmarks.

## 1 Introduction

Bayesian optimization (BO) is a powerful technique for optimizing expensive-to-evaluate black-box functions. Important applications include materials design [33], drug discovery [12], machine learning hyperparameter tuning [24], neural architecture search [16, 32], etc. BO operates by constructing a *surrogate model* for the target function, typically a Gaussian process (GP), and then using a cheap-to-evaluate *acquisition function* (AF) to guide iterative queries of the target function until a predefined budget is expended. We refer to [23] for a literature survey.

Most of the existing acquisition policies are only one-step optimal, that is, optimal if the decision horizon were one. An example is the popular *expected improvement* (EI) [20]. Such *myopic* policies only consider the immediate utility of the decision, ignoring the long-term impact of exploration. Despite the sub-optimal balancing of exploration and exploitation, they are widely used in practice due to their simplicity and computational efficiency.

When the query budget is explicitly considered, BO can be formulated as a Markov decision process (MDP), whose optimal policy maximizes the expected utility of the at the end of the decision horizon [18, 15]. However, solving the MDP is generally intractable due to the uncountable state space, uncountable action space, and potentially long decision horizon. There has been recent interest in

---

[*]Equal contribution. Work mostly done at Washington University in St. Louis for S. Jiang.

developing nonmyopic policies [11, 18, 30, 31], but these policies are often heuristic in nature or computationally expensive. A recent work known as BINOCULARS [15] achieved both efficiency and a certain degree of nonmyopia by maximizing a lower bound of the multi-step expected utility. However, a general implementation of multi-step lookahead for BO has, to our knowledge, not been attempted before.

**Main Contributions.** Our work makes progress on the intractable multi-step formulation of BO through the following methodological and empirical contributions:

- *One-shot multi-step trees.* We introduce a novel, scenario tree-based acquisition function for BO that performs an approximate, multi-step lookahead. Leveraging the reparameterization trick, we propose a way to jointly optimize all decision variables in the multi-step tree in a *one-shot* fashion, without resorting to explicit dynamic programming recursions involving nested expectations and maximizations. Our tree formulation is fully differentiable, and we compute gradients using auto-differentiation, permitting the use of gradient-based optimization.

- *Fast-fantasies and parallelism.* Our multi-step scenario tree is built by recursively sampling from the GP posterior and conditioning on the sampled function values ("fantasies"). This tree grows exponentially in size with the number of lookahead steps. While our algorithm cannot negate this reality, our novel efficient linear algebra methods for conditioning the GP model combined with a highly parallelizable implementation on accelerated hardware allows us to tackle the problem at practical wall times for moderate lookahead horizons (less than 5).

- *Improved optimization performance.* Using our method, we are able to achieve significant improvements in optimization performance over one-step EI and BINOCULARS on a range of benchmarks, while maintaining competitive wall times. To further improve scalability, we study two special cases of our general framework which are of linear growth in the lookahead horizon. We empirically show that these alternatives perform surprisingly well in practice.

We set up our problem setting in Section 2 and propose our one-shot multi-step tree approach in Section 3. We discuss how we achieve fast evaluation and optimization of these trees in Section 4. In Section 5, we show some notable instances of multi-step trees and make connections to related work in Section 6. We present our empirical results in Section 7 and conclude in Section 8.

## 2   Bayesian Optimal Policy

We consider an optimization problem

$$x^* \in \arg\max_{x \in \mathcal{X}} f(x), \tag{1}$$

where $\mathcal{X} \subset \mathbb{R}^d$ and $f(x)$ is an expensive-to-evaluate black-box function. Suppose we have collected a (possibly empty) set of initial observations $\mathcal{D}_0$ and a probabilistic surrogate model of $f$ that provides a joint distribution over outcomes $p(Y \mid X, \mathcal{D}_0)$ for all finite subsets of the design space $X \subset \mathcal{X}$. We need to reason about the locations to query the function next in order to find the maximum of $f$, given the knowledge of the remaining budget. Suppose that the location with maximum observed function value is returned at the end of the *decision horizon* $k$. A natural utility function for sequentially solving (1) is

$$u(\mathcal{D}_k) = \max_{(x,y) \in \mathcal{D}_k} y, \tag{2}$$

where $\mathcal{D}_k$ is the sequence of observations up to step $k$, defined recursively by $\mathcal{D}_i = \mathcal{D}_{i-1} \cup \{(x_i, y_i)\}$ for $i = 1, 2, \ldots, k$. Due to uncertainties in the future unobserved events, $\mathcal{D}_1, \mathcal{D}_2, \ldots, \mathcal{D}_k$ are random quantities. A policy $\boldsymbol{\pi} = (\pi_1, \pi_2, \ldots, \pi_k)$ is a collection of decision functions, where at period $i$, the function $\pi_i$ maps the dataset $\mathcal{D}_{i-1}$ to the query point $x_i$. Our objective function is $\sup_{\boldsymbol{\pi}} \mathbb{E}[u(\mathcal{D}_k^{\boldsymbol{\pi}})]$, where $\{\mathcal{D}_i^{\boldsymbol{\pi}}\}$ is the sequence of datasets generated when following $\boldsymbol{\pi}$.

For any dataset $\mathcal{D}$ and query point $x \in \mathcal{X}$, define the one-step marginal value as

$$v_1(x \mid \mathcal{D}) = \mathbb{E}_y\big[u(\mathcal{D} \cup \{(x,y)\}) - u(\mathcal{D}) \mid x, \mathcal{D}\big]. \tag{3}$$

Note that under the utility definition (2), $v_1(x \mid \mathcal{D})$ is precisely the expected improvement (EI) acquisition function [20]. It is well-known that the $k$-step problem can be decomposed via the Bellman recursion [18, 15]:

$$v_t(x \mid \mathcal{D}) = v_1(x \mid \mathcal{D}) + \mathbb{E}_y[\max_{x'} v_{t-1}(x' \mid \mathcal{D} \cup \{(x,y)\})], \tag{4}$$

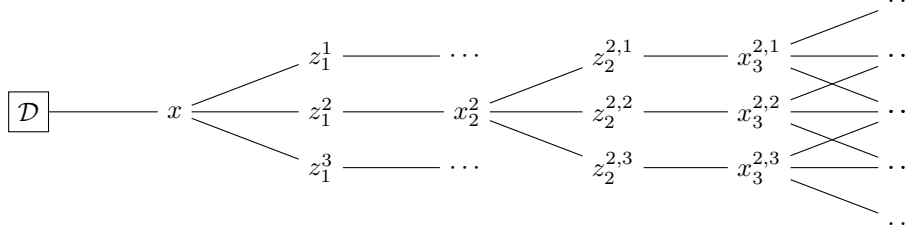

Figure 1: Illustration of the decision tree with three base samples in each stage.

for $t = 2, 3, \ldots, k$. Our $k$-step lookahead acquisition function is $v_k(x \mid \mathcal{D})$, meaning that a maximizer in $\arg\max_x v_k(x \mid \mathcal{D})$ is the recommended next point to query.

If we are allowed to evaluate multiple points $X = \{x^{(1)}, \ldots, x^{(q)}\}$ in each iteration, we replace $v$ with a batch value function $V$. For $k = 1$ and batch size $|X| = q$, we have

$$V_1^q(X \mid \mathcal{D}) = \mathbb{E}_{y^{(1)}, \ldots, y^{(q)}} \left[ u(\mathcal{D} \cup \{(x^{(1)}, y^{(1)}), \ldots, (x^{(q)}, y^{(q)})\}) - u(\mathcal{D}) \mid X, \mathcal{D} \right],$$

which under the utility definition (2) is known as $q$-EI in the literature [10, 26]. For general $k$, $V_k$ is the exact analogue of (4); we capitalize $v$ and $x$ to indicate expected value of a batch of points. While we only consider the fully adaptive setting ($q = 1$) in this paper, we will make use of the batch policy for approximation.

## 3 One-Shot Optimization of Multi-Step Trees

In this section, we describe our multi-step lookahead acquisition function, a differentiable, tree-based approximation to $v_k(x \mid \mathcal{D})$. We then propose a one-shot optimization technique for effectively optimizing the acquisition function and extracting a first-stage decision.

### 3.1 Multi-Step Trees

Solving the $k$-step problem requires recursive maximization and integration over continuous domains:

$$v_k(x \mid \mathcal{D}) = v_1(x \mid \mathcal{D}) + \mathbb{E}_y \left[ \max_{x_2} \left\{ v_1(x_2 \mid \mathcal{D}_1) + \mathbb{E}_{y_2} \left[ \max_{x_3} \left\{ v_1(x_3 \mid \mathcal{D}_2) + \cdots \right]. \right. \right. \tag{5}$$

Since under a GP surrogate, these nested expectations are analytically intractable (except the last step for EI), we must resort to numerical integration. If we use Monte Carlo integration, this essentially means building a discrete scenario tree (Figure 1), where each branch in a node corresponds to a particular *fantasized* outcome drawn from the model posterior, and then averaging across scenarios. Letting $m_t$, $t = 1, \ldots, k - 1$ denote the number of *fantasy samples* from the posterior in step $t$, we have the approximation

$$\bar{v}_k(x \mid \mathcal{D}) = v_1(x \mid \mathcal{D}) + \frac{1}{m_1} \sum_{j_1=1}^{m_1} \left[ \max_{x_2} \left\{ v_1(x_2 \mid \mathcal{D}_1^{j_1}) + \frac{1}{m_2} \sum_{j_2=1}^{m_2} \left[ \max_{x_3} \left\{ v_1(x_3 \mid \mathcal{D}_2^{j_1 j_2}) + \cdots \right], \right. \right. \right.$$

where $\mathcal{D}_1^{j_1} = \mathcal{D} \cup \{(x, y^{j_1})\}$, $\mathcal{D}_t^{j_1 \cdots j_t} = \mathcal{D}_{t-1}^{j_1 \cdots j_{t-1}} \cup \{(x_t^{j_1 \cdots j_{t-1}}, y_t^{j_1 \cdots j_t})\}$, with fantasy samples $y_t^{j_1 \cdots j_t} \sim p(y_t \mid x_t^{j_1 \cdots j_t}, \mathcal{D}_{t-1}^{j_1 \cdots j_{t-1}})$. As the distribution of the fantasy samples depends on the query locations $x, x_1, x_2, \ldots$, we cannot directly optimize $\bar{v}_k(x \mid \mathcal{D})$. To make $\bar{v}_k(x \mid \mathcal{D})$ amenable to optimization, we leverage the re-parameterization trick [17, 28] to write $y = h_{\mathcal{D}}(x, z)$, where $h_{\mathcal{D}}$ is a deterministic function and $z$ is a random variable independent of both $x$ and $\mathcal{D}$. Specifically, for a GP posterior, we have $h_{\mathcal{D}}(x, z) = \mu_{\mathcal{D}}(x) + L_{\mathcal{D}}(x)z$, where $\mu_{\mathcal{D}}(x)$ is the posterior mean, $L_{\mathcal{D}}(x)$ is a root decomposition of the posterior covariance $\Sigma_{\mathcal{D}}(x)$ such that $L_{\mathcal{D}}(x)L_{\mathcal{D}}^T(x) = \Sigma_{\mathcal{D}}(x)$, and $z \sim \mathcal{N}(0, I)$. Since a GP conditioned on additional samples remains a GP, we can perform a similar re-parameterization for every dataset $\mathcal{D}_t^{j_1 \cdots j_t}$ in the tree. We refer to the $z$'s as *base samples*.

### 3.2 One-Shot Optimization

Despite re-parameterizing $\bar{v}_k(x \mid \mathcal{D})$ using base samples, it still involves nested maximization steps. Particularly when each optimization must be performed numerically using sequential approaches (as

is the case when auto-differentiation and gradient-based methods are used), this becomes cumbersome. Observe that conditional on the base samples, $\bar{v}_k$ is a *deterministic* function of the decision variables.

**Proposition 1.** *Fix a set of base samples and consider $\bar{v}_k(x \,|\, \mathcal{D})$. Let $x_t^{j_1\dots j_{t-1}}$ be an instance of $x_t$ for each realization of $\mathcal{D}_{t-1}^{j_1\dots j_{t-1}}$ and let*

$$x^*, \mathbf{x}_2^*, \mathbf{x}_3^*, \dots, \mathbf{x}_k^* = \operatorname*{arg\,max}_{x, \mathbf{x}_2, \mathbf{x}_3, \dots, \mathbf{x}_k} \left\{ v_1(x \,|\, \mathcal{D}) + \frac{1}{m_1} \sum_{j_1=1}^{m_1} v_1(x_2^{j_1} \,|\, \mathcal{D}_1^{j_1}) + \dots + \right.$$
$$\left. \frac{1}{\prod_{\ell=1}^{k-1} m_\ell} \sum_{j_1=1}^{m_1} \dots \sum_{j_{k-1}=1}^{m_{k-1}} v_1(x_k^{j_1\dots j_{k-1}} \,|\, \mathcal{D}_{k-1}^{j_1\dots j_{k-1}}) \right\}, \quad (6)$$

*where we compactly represent $\mathbf{x}_2 = \{x_2^{j_1}\}_{j_1=1\dots m_1}$, $\mathbf{x}_3 = \{x_3^{j_1 j_2}\}_{j_1=1\dots m_1, j_2=1\dots m_2}$, and so on. Then, $x^* = \arg\max_x \bar{v}_k(x \,|\, \mathcal{D})$.*

Proposition 1 suggests that rather than solving a nested optimization problem, we can solve a joint optimization problem of higher dimension and subsequently extract the optimizer. We call this the *one-shot multi-step* approach. A single-stage version of this was used in [1] for optimizing the Knowledge Gradient (KG) acquisition function [29], which also has a nested maximization (of the posterior mean). Here we generalize the idea to its full extent for efficient multi-step BO. We use a perturbed version of the solution from the last iteration to warm-start the optimization of (6); technical details can be found in Appendix D. We will show that this can dramatically improve the performance in practice.

## 4    Fast, Differentiable, Multi-Step Fantasization

Computing the one-shot objective (6) requires us to repeatedly condition the model on the fantasy samples as we traverse the tree to deeper levels. Our ability to solve multi-step lookahead problems efficiently is made feasible by linear algebra insights and careful use of efficient batched computation on modern parallelizable hardware. Typically, conditioning a GP on additional data in a computationally efficient fashion is done by performing rank-1 updates to the Cholesky decomposition of the input covariance matrix. In this paper, we develop a related approach, which we call *multi-step fast fantasies*, in order to efficiently construct fantasy models for GPyTorch [8] GP models representing the full lookahead tree. A core ingredient of this approach is a novel linear algebra method for efficiently updating GPyTorch's LOVE caches [22] for posterior inference in each step.

### 4.1    Background: Lanczos Variance Estimates

We start by providing a brief review of the main concepts for the Lanczos Variance Estimates (LOVE) as introduced in [22]. The GP predictive covariance between two test points $x_i^*$ and $x_j^*$ is given by:

$$k_{f|\mathcal{D}}(x_i^*, x_j^*) = k_{x_i^* x_j^*} - \mathbf{k}_{Xx_i^*}^\top (K_{XX} + \Sigma)^{-1} \mathbf{k}_{Xx_j^*},$$

$X = (x_1, \dots, x_n)$ is the set of training points, $K_{XX}$ is the kernel matrix at $X$, and $\Sigma$ is the noise covariance.[2] LOVE achieves fast (co-)variances by decomposing $K_{XX} + \Sigma = RR^\top$ in $\mathcal{O}(r\nu(K_{XX}))$ time, where $R \in \mathbb{R}^{n \times r}$ and $\nu(K_{XX})$ is the time complexity of a matrix vector multiplication $K_{XX}v$. This allows us to compute the second term of the predictive covariance as:

$$k_{f|\mathcal{D}}(x_i^*, x_j^*) = k_{x_i^* x_j^*} - \mathbf{k}_{Xx_i^*}^\top R^{-\top} R^{-1} \mathbf{k}_{Xx_j^*},$$

where $R^{-1}$ denotes a pseudoinverse if $R$ is low-rank.[3] The main operation to perform is decomposing $\tilde{K}_{XX} = RR^\top$, where $\tilde{K}_{XX} := K_{XX} + \Sigma \in \mathbb{R}^{n \times n}$. Computing this decomposition can be done from scratch in $\mathcal{O}(nr^2)$ time. After forming $R$, additional $\mathcal{O}(nr^2)$ time is required to perform a QR decomposition of $R$ so that a linear least squares systems can be solved efficiently (i.e., approximate $R^{-1}$). $R$ and its QR decomposition are referred to as the LOVE *cache*.

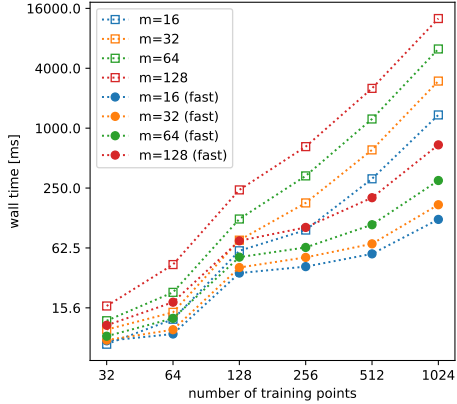

Figure 2: CPU times for constructing fantasy model and evaluating its posterior at a single point (variance negligible relative to the mean).

**Algorithm 1:** Multi-Step Tree Evaluation

---

$\text{VALUE}(\mathcal{M}_t, \boldsymbol{X}_t, \mathcal{D}_{t-1})$:
&emsp;$y_{t-1}^* = \max_{(x,y) \in \mathcal{D}_{t-1}} y$
&emsp;**return** $\mathbb{E}_{y \sim \mathcal{M}_t(\boldsymbol{X}_t)} \left[ (y - y_{t-1}^*)^+ \right]$
$\text{STEP}(\alpha_t, \mathcal{M}_t, \boldsymbol{X}_{t:k}, \boldsymbol{Z}_{t:k}, \mathcal{D}_{t-1})$:
&emsp;$\alpha_{t+1} = \alpha_t + \text{VALUE}(\mathcal{M}_t, \boldsymbol{X}_t, \mathcal{D}_{t-1})$
&emsp;**if** $t = k - 1$ **then**
&emsp;&emsp;**return** $\alpha_{t+1}$
&emsp;$Y_t = \text{CORRELATE}(\mathcal{M}_t(\boldsymbol{X}_t), \boldsymbol{Z}_t)$
&emsp;$\mathcal{M}_{t+1} = \text{FANTASIZE}(\mathcal{M}_t, \boldsymbol{X}_t, \boldsymbol{Y}_t)$
&emsp;$\mathcal{D}_t = \mathcal{D}_{t-1} \cup \{(\boldsymbol{X}_t, \boldsymbol{Y}_t)\}$
&emsp;**return** $\text{STEP}(\alpha_{t+1}, \mathcal{M}_{t+1}, \boldsymbol{X}_{t+1:k}, \boldsymbol{Z}_{t+1:k}, \mathcal{D}_t)$

---

Figure 3: The recursive procedure for evaluating multi-step trees by repeatedly sampling from the posterior (CORRELATE), conditioning (FANTASIZE), and evaluating stage-values (VALUE).

## 4.2 Fast Cache Updates

If $R$ were a full Cholesky decomposition of $\tilde{K}_{XX}$, it could be updated in $\mathcal{O}(n^2)$ time using well-known procedures for rank 1 updates to Cholesky decompositions. This is advantageous, because computing the Cholesky decomposition requires $\mathcal{O}(n^3)$ time. However, for dense matrices, the LOVE cache requires only $\mathcal{O}(n^2 r)$ time to compute from scratch. Therefore, an update routine is only efficient if it can be performed in less (i.e., $o(n^2)$) time. Updating the LOVE caches is in particular complicated by the fact that $R$ is not necessarily triangular (or even square). Therefore, unlike with a Cholesky decomposition, updating $R$ itself in quadratic time is not sufficient, as recomputing a QR decomposition of $R$ to update the pseudoinverse $R^\dagger$ would itself take quadratic time. In the Appendix E, we demonstrate that the following proposition is true:

**Proposition 2.** *Suppose $(K_{XX} + \Sigma)^{-1}$ has been decomposed using LOVE into $R^{-\top} R^{-1}$, with $R^{-1} \in \mathbb{R}^{n \times r}$. Suppose we wish to augment $X$ with $q$ data points, thereby augmenting $K_{XX}$ with $q$ rows and columns, yielding $K_{\hat{X}\hat{X}}$. A rank $r + q$ decomposition $\hat{R}^{-1}$ of the inverse, $\hat{R}^{-\top} \hat{R}^{-1} \approx (K_{\hat{X}\hat{X}} + \Sigma)^{-1}$, can be computed from $R$ in $\mathcal{O}(nrq)$ time.*

## 4.3 Multi-Step Fantasies and Scalability

Our other core insight is that the different levels of the lookahead tree can be represented by the batch dimensions of batched GP models; this allows us to exclusively use batched linear algebra tensor operations that heavily exploit parallelization and hardware acceleration for our computations. This optimized implementation is crucial in order to scale to non-trivial multi-step problems. Algorithm 1 shows our recursive implementation of (6).[4] Using reparameterization, we retain the dependence of the value functions in all stages on $x, \mathbf{x}_2, \ldots, \mathbf{x}_k$, and can auto-differentiate through Algorithm 1.

Figure 2 compares the overall wall time (on a logarithmic scale) for constructing fantasy models and performing posterior inference, for both standard and fast fantasy implementations. Fast fantasies achieve substantial speedups with growing model size and number of fantasies (up to 16x for $n = 1024$, $m = 128$). In Appendix E, we show that computations on a GPU are substantially faster for larger models and number of fantasies.

## 5 Special Instances of Multi-Step Trees

The general one-shot optimization problem is of dimension $d + d \sum_{t=1}^{k} \prod_{i=1}^{t} m_i$, which grows exponentially in the lookahead horizon $k$. Therefore, in practice we are limited to relatively small horizons $k$ (nevertheless, we are able to show experimental results for up to $k = 4$, which has never been considered in the literature). In this section, we describe two alternative approaches that have dimension only linear in $k$ and have the potential to be even more scalable.

**Multi-Step Path.** Suppose only a single path is allowed in each subtree rooted at each fantasy sample $y^{j_1}, j_1 = 1, \ldots, m_1$, that is, let $m_t = 1$ for $t \geq 2$, then the number of variables is linear in $k$ and $m_1$. An even more extreme case is further setting $m_1 = 1$, that is, we assume there is only one possible future path. When Gauss-Hermite (GH) quadrature rules are used (one sample is exactly the mean of the Gaussian), this approach has a strong connection with *certainty equivalent control* [2], an approximation technique for solving optimal control problems. It also relates to some of the notable batch construction heuristics such as Kriging Believer [10], or GP-BUCB [4], where one fantasizes (or "hallucinates" as is called in [4]) the posterior mean and continues simulating future steps as if it were the actual observed value. We will see that this degenerate tree can work surprisingly well in practice.

**Non-Adaptive Approximation.** If we approximate the adaptive decisions after the current step by non-adaptive decisions, the *one-shot* optimization would be

$$\max_{x, X^{(1)}, \ldots, X^{(m_1)}} v_1(x \mid \mathcal{D}) + \frac{1}{m_1} \sum_{i=1}^{m_1} V_1^{k-1}(X^{(i)} \mid \mathcal{D}_1^{(i)}), \tag{7}$$

where we replaced the adaptive value function $v_{k-1}$ by the one-step batch value function $V_1^{k-1}$ with batch size $k - 1$, i.e., $|X| = k - 1$. The dimension of (7) is $d + m_1(k-1)d$. Since non-adaptive expected utility is no greater than the adaptive expected utility, (7) is a lower bound of the adaptive expected utility. Such non-adaptive approximation is actually a proven idea for *efficient nonmyopic search* (ENS) [13, 14], a problem setting closely related to BO. We refer to (7) as ENO, for *efficient nonmyopic optimization*. See Appendix B for further discussions of these two special instances.

## 6 Related Work

While we consider a general multi-step lookahead setup, there are several earlier attempts on two-step lookahead BO [21, 9]. The most closely related work is a recent development that made gradient-based optimization of two-step EI possible [30]. In their approach, maximizers of the second-stage value functions conditioned on each $y$ sample of the first stage are first identified, and then substituted back. If certain conditions are satisfied, this function is differentiable and admits unbiased stochastic gradient estimation (via the envelope theorem). This method relies on the assumption that the maximizers of the second-stage value functions are *global optima*. This assumption can be unrealistic, especially for high-dimensional problems. Any violation of this assumption would result in discontinuity of the objective, and differentiation would be problematic.

*Rollout* is a classical approach in approximate dynamic programming [3, 25] and adapted to BO by [18, 31]. However, rollout estimation of the expected utility is only a lower bound of the true multi-step expected utility, because a *base policy* is evaluated instead of the optimal policy. Another notable nonmyopic BO policy is GLASSES [11], which also uses a batch policy to approximate future steps. Unlike ENO, GLASSES uses a heuristic batch instead of the optimal one, and, perhaps more crucially, its batch is not adaptive to the sample values of the first stage. All three methods discussed above share similar repeated, nested optimization procedures for each evaluation of the acquisition function and hence are very expensive to optimize.

Recently [15] proposed an efficient nonmyopic approach called BINOCULARS, where a point from the optimal batch is selected at random. This heuristic is justified by the fact that any point in the batch maximizes a lower bound of the true expected utility that is tighter than GLASSES. We summarize all the methods discussed in this paper in Table 1, in which we also present comparisons of the tightness of the lower bounds. Note that "multi-step path" can be considered a noisy version of "multi-step".

Table 1: Summary and comparison of the nonmyopic approaches discussed in this paper. Notation: for GLASSES, $X_g$ is a heuristically constructed batch; note it does not depend on $y$; for rollout, $r_1(x \mid \mathcal{D}) = v_1(x \mid \mathcal{D})$, and we assume $\pi(\mathcal{D}_1) = \arg\max_x v_1(x \mid \mathcal{D}_1)$ is the base policy for a meaningful comparison. Recall $\mathcal{D}_1 = \mathcal{D} \cup \{(x, y)\}$. The relationships are due to (1) non-adaptive expected utility is a lower bound on adaptive expected utility, and (2) $\mathbb{E}[\max \cdot] \geq \max \mathbb{E}[\cdot]$.

| Method | Acquisition Function |
|---|---|
| multi-step (ours) | $v_1(x \mid \mathcal{D}) + \mathbb{E}_y[\max_{x'} v_{k-1}(x' \mid \mathcal{D}_1)]$ |
| ENO (ours) | $v_1(x \mid \mathcal{D}) + \mathbb{E}_y[\max_X V_1^{k-1}(X \mid \mathcal{D}_1)]$ |
| BINOCULARS [15] | $v_1(x \mid \mathcal{D}) + \max_X \mathbb{E}_y[V_1^{k-1}(X \mid \mathcal{D}_1)]$ |
| GLASSES [11] | $v_1(x \mid \mathcal{D}) + \mathbb{E}_y[V_1^{k-1}(X_g \mid \mathcal{D}_1)]$ |
| rollout [18] | $r_k(x \mid \mathcal{D}) = r_1(x \mid \mathcal{D}) + \mathbb{E}_y[r_{k-1}(\pi(\mathcal{D}_1) \mid \mathcal{D}_1)]$ |
| two-step [30] | $v_1(x \mid \mathcal{D}) + \mathbb{E}_y[\max_{x'} v_1(x' \mid \mathcal{D}_1)]$ |
| one-step [**?** ] | $v_1(x \mid \mathcal{D}) + 0$ |
| relationships (when $k \geq 2$) | multi-step $\geq$ ENO $\geq$ BINOCULARS $\geq$ GLASSES $\geq$ one-step; multi-step $\geq$ rollout $\geq$ two-step $\geq$ one-step;  ENO $\geq$ two-step. |

Table 2: Average GAP and time (seconds) per iteration. We run 100 repeats for each method and each function with $2d$ random initial observations of the function. Bold: best, blue: not significantly worse than the best under paired one-sided sign rank test with $\alpha = 0.05$.

| | EI | ETS | 12.EI.s | 2-step | 3-step | 4-step | 4-path | 12-ENO |
|---|---|---|---|---|---|---|---|---|
| eggholder | 0.627 | 0.647 | **0.736** | 0.478 | 0.536 | 0.577 | 0.567 | 0.661 |
| dropwave | 0.429 | 0.585 | 0.606 | 0.545 | 0.600 | 0.635 | **0.731** | 0.673 |
| shubert | 0.376 | *0.487* | *0.515* | 0.476 | *0.507* | **0.562** | *0.560* | *0.494* |
| rastrigin4 | 0.816 | 0.495 | 0.790 | **0.851** | *0.821* | *0.826* | *0.837* | *0.837* |
| ackley2 | 0.808 | 0.856 | *0.902* | *0.870* | *0.895* | *0.888* | **0.931** | 0.847 |
| ackley5 | 0.576 | 0.516 | 0.703 | 0.786 | 0.793 | 0.804 | **0.875** | *0.856* |
| bukin | 0.841 | 0.843 | 0.842 | **0.862** | *0.862* | *0.861* | *0.852* | 0.836 |
| shekel5 | 0.349 | 0.132 | 0.496 | *0.827* | **0.856** | *0.847* | 0.718 | 0.799 |
| shekel7 | 0.363 | 0.159 | 0.506 | *0.825* | *0.850* | 0.775 | 0.776 | **0.866** |
| Avg. GAP | 0.576 | 0.524 | 0.677 | 0.725 | *0.747* | *0.753* | *0.761* | **0.763** |
| Avg. time | 1.157 | 1949. | 25.74 | 7.163 | 39.53 | 197.7 | 17.50 | 15.61 |

## 7 Experiments

We follow the experimental setting of [15], and test our algorithms using the same set of synthetic and real benchmark functions. For brevity, we only present the results on the synthetic benchmarks here; additional results are given in Appendix F. All algorithms are implemented in `BoTorch` [1], and we use a GP with a constant mean and a Matérn $5/2$ ARD kernel for BO. GP hyperparameters are re-estimated by maximizing the evidence after each iteration. For each experiment, we start with $2d$ random observations, and perform $20d$ iterations of BO; 100 experiments are repeated for each function and each method. We measure performance with GAP $= (y_i - y_0)/(y^* - y_0)$. All experiments are run on CPU Linux machines; each experiment only uses one core.

[15] used nine "hard" synthetic functions, motivated by the argument that the advantage of nonmyopic policies are more evident when the function is hard to optimize. We follow their work and use the same nine functions. [15] thoroughly compares BINOCULARS with some well known nonmyopic baselines such as rollout and GLASSES and demonstrates superior results, so we will focus on comparing with EI, BINOCULARS and the "envelope two-step" (ETS) method [30]. We choose the best reported variant, 12.EI.s, for BINOCULARS on these functions [15], i.e., first compute an optimal batch of 12 points (maximize $q$-EI), then sample a point from it weighted by the individual EI values. All details can be found in our accompanying code submission.

We use the following nomenclature: "$k$-step" means $k$-step lookahead ($k = 2, 3, 4$) with number of GH samples $m_1 = 10, m_2 = 5, m_3 = 3$. These numbers are heuristically chosen to favor more

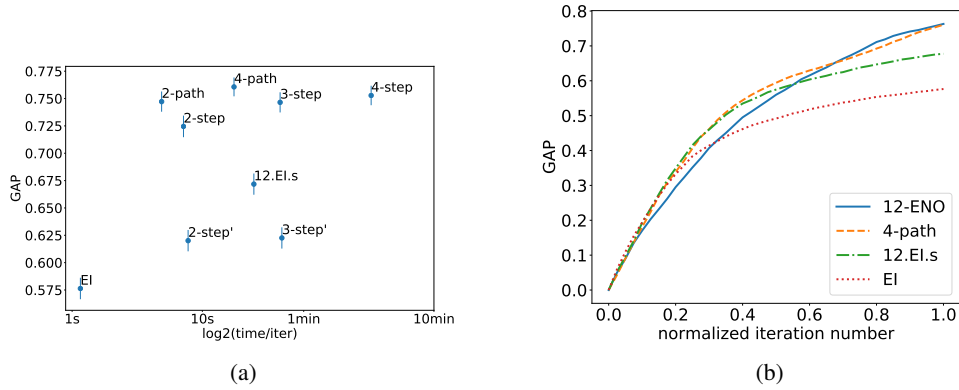

(a)                                                                    (b)

Figure 4: (a) Aggregated GAP with error bars vs. time per iteration, averaged over the nine synthetic functions by 100 repeats. (b) Aggregated optimization trace: GAP versus number of iterations, normalized into 0-1.

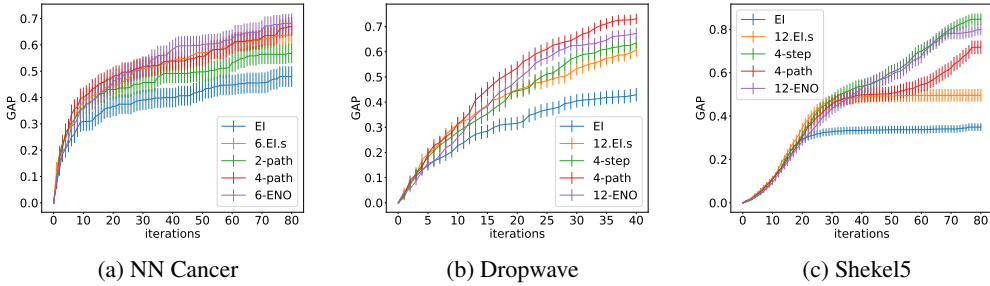

(a) NN Cancer                          (b) Dropwave                          (c) Shekel5

Figure 5: GAP vs. #iterations for three benchmarks: tuning a classification neural network on a breast cancer dataset and two synthetic functions.

accuracy in earlier stages. "$k$-path" is the multi-step path variant that uses only one sample for each stage. "$k$-ENO" is ENO using non-adaptive approximation of the future $k-1$ steps with a $q$-EI batch with $q = k - 1$. The average terminal GAP and the average time (in seconds) per iteration for all methods are presented in Table 2. Figure 4 gives results for individual methods when averaged across all functions. Figure 5 shows the full learning curves for three selected benchmarks problems; additional results are given in Appendices F and G. Following existing research [18, 30], the results above use GH quadrature to generate samples for approximating the expectation in each stage, with the number of samples fixed heuristically. In Table 3, we compare two strategies for approximating the expectation, Gauss-Hermite quadrature (GH) and quasi-Monte-Carlo (MC), while varying number of samples in the tree, on BO performance for a fixed optimization budget.[5] In Figure 6, we show the average time per iteration vs. the number of samples used in the tree. Some entries are omitted from the table and plots to aid with the presentation. We give more details and takeaways below.

- *Baselines.* First, we see from Table 2 ETS outperforms EI on $5/9$ of the functions, but on average is worse than EI, especially on the two Shekel functions, and the average time per iteration is over 30min. Our results for 12.EI.s and EI closely match those reported in [15].

- *One-shot multi-step.* We see from Figure 4(a) that *our 2,3,4-step lookahead methods outperform all baselines by a large margin*. We also see diminishing returns with increasing horizon, with only a minor improvement beyond 3-step. It is not clear whether this is due to ineffective optimization of the increasingly complex multi-step objective, or if additional lookahead means increasing reliance on the model being accurate, which is often not the case in practice [31].

- *One-shot pseudo multi-step.* Note that the average time per iteration grows exponentially with the lookahead horizon if we use multiple fantasy samples for each stage. In Figure 4(a), we see

Table 3: Average GAP results while varying the sampling method (Gauss-Hermite (GH) or quasi-Monte-Carlo (MC)) and number of samples $m_1$ on the nine synthetic functions with 100 repeats each. Underline means significantly worse (signed rank test $\alpha = 0.05$).

| $m_1$ | 2-step | | 3-step | | 4-step | |
|---|---|---|---|---|---|---|
| | GH | MC | GH | MC | GH | MC |
| 1 | 0.747 | **0.754** | 0.754 | **0.784** | 0.757 | **0.774** |
| 2 | **0.751** | 0.728 | 0.762 | **0.768** | **0.777** | 0.769 |
| 4 | **0.766** | 0.737 | **0.767** | 0.748 | **0.773** | 0.768 |
| 8 | **0.738** | 0.720 | **0.750** | 0.736 | **0.762** | 0.735 |
| 16 | **0.734** | 0.724 | 0.750 | **0.752** | – | – |
| 32 | **0.745** | 0.733 | **0.732** | 0.725 | – | – |

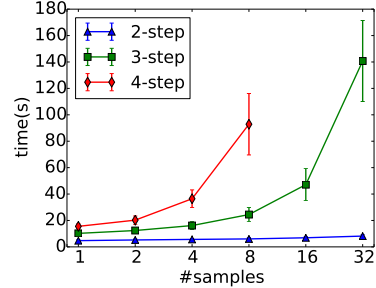

Figure 6: Average time per iteration vs. number of samples $m_1$.

2- and 4-path produce similar results in significantly less time, suggesting that a noisy version of the multi-step tree is a reasonable alternative. The average GAP and time for 2,3,4-path are: GAP = 0.747, 0.750, 0.761, time = 4.87s, 10.4s, 17.5s. We also run 12-ENO with $k = 12$, which is chosen to match 12.EI.s. The GAP and time per iteration are very close to 4-path. Interestingly, Figure 4(b) shows a pronounced *nonmyopic behavior* in 12-ENO that one might expect from a long lookahead horizon: 12-ENO first underperforms, but eventually outperforms the other methods. Similar behavior has been consistently observed in efficient nonmyopic active search [13, 14]. Results for individual functions are given in Appendix G. Note that the pseudo multi-step objective is of lower dimension than its "full tree" counterpart and hence easier to optimize — this may partly contribute to its effectiveness.

- *Warm-starting.* We use a perturbed version of the solution from previous iteration to warm-start the optimization of the one-shot objective. In Figure 4(a), we see that the BO performance of 2- and 3-step without warm-start, as indicated by 2-step' and 3-step', is substantially worse (but still significantly better than EI). These results suggest that further study on warm-start techniques for optimizing BO acquisition functions would be valuable.

- GH *vs.* MC *and number of samples*. In Table 3, we show the performance of 2-, 3- and 4-step with GH and MC sampling, varying number of samples by $m_1 = 1, 2, 4, 8, 16, 32$, $m_2 = \max(1, m_1/2)$, and $m_3 = \max(1, m_2/2)$.[6] First, we notice that using more samples is exponentially more expensive, but not necessarily better. Although we might expect that a more accurate approximation of the expected utility leads to better BO performance, there are a number of competing factors, including (1) increased difficulty of acquisition function optimization, (2) model mis-specification [31], and (3) the rolling-horizon implementation of the approach. Second, we observe that GH generally performs slightly better than MC, with an exception when we only use one sample (i.e., multi-step path), where MC consistently shows better results. In fact, 3-path with one MC sample achieved the best average result (0.7841) on these set of benchmarks. This could partially be explained by the fact that multi-path is, in expectation, an upper bound of the true expected utility, as explained in Appendix B. These points deserve further exploration in future work.

## 8   Conclusion

General multi-step lookahead Bayesian optimization is a notoriously hard problem. We provide the first efficient implementation based on a simple idea: jointly optimize all decision variables of a multi-step scenario tree in *one-shot*, instead of naively computing the nested expectation and maximization. Our implementation relies on fast, differentiable fantasization, highly batched and vectorized recursive sampling and conditioning of Gaussian processes, and auto-differentiation. Results on a wide range of benchmarks demonstrate its high efficiency and optimization performance. We also find that two special cases, multi-step path and non-adaptive approximation of future decisions, work as well, if not better, while requiring fewer computational resources. An interesting future endeavor is to investigate the application of our framework to other problems, such as Bayesian quadrature [15].

## Broader Impact

The central concern of this investigation is Bayesian optimization of an expensive-to-evaluate objective function. As is standard in this body of literature, our proposed algorithms make minimal assumptions about the objective, effectively treating it as a "black box." This abstraction is mathematically convenient but ignores ethical issues related to the chosen objective. Traditionally, Bayesian optimization has been used for a variety of applications, including materials design and drug discovery [7], and could have future applications to algorithmic fairness. We anticipate that our methods will be utilized in these reasonable applications, but there is nothing inherent to this work, and Bayesian optimization as a field more broadly, that preclude the possibility of optimizing a nefarious or at least ethically complicated objective.

## Acknowledgement

Garnett is supported by the National Science Foundation (NSF) under award numbers IIS–1939677, OAC–1940224, and IIS–1845434.

## Footnotes

[2]Typically, $\Sigma = \sigma^2 I$, but other formulations, including heteroskedastic noise models, are also compatible with fast fantasies described here.

[3]Additional approximations can be performed when using Spectral Kernel Interpolation (SKI), which result in constant time predictive covariances. For simplicity, we only detail the case of exact GPs here.

[4]Here $\mathcal{M}_t(\boldsymbol{x}_t)$ denotes the posterior of the model $\mathcal{M}_t$ evaluate at $\boldsymbol{x}_t$, and $\boldsymbol{X}_{t:k} := \{\boldsymbol{x}_i\}_{i=t}^k$ and $\boldsymbol{Z}_{t:k} := \{\boldsymbol{z}_i\}_{i=t}^k$ are collections of decision variables and base samples for lookahead steps $t$ through $k$, respectively. CORRELATE$(\mathcal{M}_t(\boldsymbol{X}_t), \boldsymbol{Z}_t)$ generates fantasy samples by correlating the base samples $\boldsymbol{Z}_t$ via the model posterior $\mathcal{M}_t(\boldsymbol{X}_t)$, and FANTASIZE$(\mathcal{M}_t, \boldsymbol{X}_t, \boldsymbol{Y}_t)$ produces a new fantasy model (with an additional batch dimension) by conditioning on the fantasized observations. To compute $k$-step one-shot lookahead conditional on base samples $\boldsymbol{Z}_{0:k}$ at decision variables $\boldsymbol{X}_{0:k}$, we simply need to call STEP$(0, \mathcal{M}, \boldsymbol{X}_{0:k}, \boldsymbol{Z}_{0:k}, \mathcal{D})$.

[5]These experiments were run in response to the questions raised by the anonymous reviewers. Due to time constraints, we set the budget of 300 function evaluations for optimizing the one-shot objective.

[6]We did not run 16- and 32-sample for 4-step due to their high computational requirements.

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
