[Supplementary Material]

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

_1 \ldots j_t} = \mathcal{D}_{t-1}^{j_1 \ldots j_{t-1}} \cup \{(x_t^{j_1 \ldots j_{t-1}}, y_t^{j_1 \ldots j_t})\}$, with fantasy samples $y_t^{j_1 \ldots j_t} \sim p(y_t \mid x_t^{j_1 \ldots j_t}, \mathcal{D}_{t-1}^{j_1 \ldots j_{t-1}})$. As the distribution of the fantasy samples depends on the query locations $x, x_1, x_2, \ldots$, we cannot directly optimize $\bar{v}_k(x \mid \mathcal{D})$. To make $\bar{v}_k(x \mid \mathcal{D})$ amenable to optimization, we leverage the re-parameterization trick [17, 28] to write $y = h_{\mathcal{D}}(x, z)$, where $h_{\mathcal{D}}$ is a deterministic function and $z$ is a random variable independent of both $x$ and $\mathcal{D}$. Specifically, for a GP posterior, we have $h_{\mathcal{D}}(x, z) = \mu_{\mathcal{D}}(x) + L_{\mathcal{D}}(x)z$, where $\mu_{\mathcal{D}}(x)$ is the posterior mean, $L_{\mathcal{D}}(x)$ is a root decomposition of the posterior covariance $\Sigma_{\mathcal{D}}(x)$ such that $L_{\mathcal{D}}(x)L_{\mathcal{D}}^T(x) = \Sigma_{\mathcal{D}}(x)$, and $z \sim \mathcal{N}(0, I)$. Since a GP conditioned on additional samples remains a GP, we can perform a similar re-parameterization for every dataset $\mathcal{D}_t^{j_1 \ldots j_t}$ in the tree. We refer to the $z$'s as *base samples*.

### 3.2 One-Shot Optimization

Despite re-parameterizing $\bar{v}_k(x \mid \mathcal{D})$ using base samples, it still involves nested maximization steps. Particularly when each optimization must be performed numerically using sequential approaches (as

is the case when auto-differentiation and gradient-based methods are used), this becomes cumbersome. Observe that conditional on the base samples, $\bar{v}_k$ is a *deterministic* function of the decision variables.

**Proposition 1.** *Fix a set of base samples and consider $\bar{v}_k(x \,|\, \mathcal{D})$. Let $x_t^{j_1 \cdots j_{t-1}}$ be an instance of $x_t$ for each realization of $\mathcal{D}_{t-1}^{j_1 \cdots j_{t-1}}$ and let*

$$x^*, \mathbf{x}_2^*, \mathbf{x}_3^*, \ldots, \mathbf{x}_k^* = \underset{x, \mathbf{x}_2, \mathbf{x}_3, \ldots, \mathbf{x}_k}{\arg\max} \left\{ v_1(x \,|\, \mathcal{D}) + \frac{1}{m_1} \sum_{j_1=1}^{m_1} v_1(x_2^{j_1} \,|\, \mathcal{D}_1^{j_1}) + \cdots + \right.$$
$$\left. \frac{1}{\prod_{\ell=1}^{k-1} m_\ell} \sum_{j_1=1}^{m_1} \cdots \sum_{j_{k-1}=1}^{m_{k-1}} v_1(x_k^{j_1 \cdots j_{k-1}} \,|\, \mathcal{D}_{k-1}^{j_1 \cdots j_{k-1}}) \right\}, \quad (6)$$

*where we compactly represent $\mathbf{x}_2 = \{x_2^{j_1}\}_{j_1=1 \ldots m_1}$, $\mathbf{x}_3 = \{x_3^{j_1 j_2}\}_{j_1=1 \ldots m_1, j_2=1 \ldots m_2}$, and so on. Then, $x^* = \arg\max_x \bar{v}_k(x \,|\, \mathcal{D})$.*

Proposition 1 suggests that rather than solving a nested optimization problem, we can solve a joint optimization problem of higher dimension and subsequently extract the optimizer. We call this the *one-shot multi-step* approach. A single-stage version of this was used in [1] for optimizing the Knowledge Gradient (KG) acquisition function [29], which also has a nested maximization (of the posterior mean). Here we generalize the idea to its full extent for efficient multi-step BO. We use a perturbed version of the solution from the last iteration to warm-start the optimization of (6); technical details can be found in Appendix D. We will show that this can dramatically improve the performance in practice.

## 4 Fast, Differentiable, Multi-Step Fantasization

Computing the one-shot objective (6) requires us to repeatedly condition the model on the fantasy samples as we traverse the tree to deeper levels. Our ability to solve multi-step lookahead problems efficiently is made feasible by linear algebra insights and careful use of efficient batched computation on modern parallelizable hardware. Typically, conditioning a GP on additional data in a computationally efficient fashion is done by performing rank-1 updates to the Cholesky decomposition of the input covariance matrix. In this paper, we develop a related approach, which we call *multi-step fast fantasies*, in order to efficiently construct fantasy models for GPyTorch [8] GP models representing the full lookahead tree. A core ingredient of this approach is a novel linear algebra method for efficiently updating GPyTorch's LOVE caches [22] for posterior inference in each step.

### 4.1 Background: Lanczos Variance Estimates

We start by providing a brief review of the main concepts for the Lanczos Variance Estimates (LOVE) as introduced in [22]. The GP predictive covariance between two test points $x_i^*$ and $x_j^*$ is given by:

$$k_{f|\mathcal{D}}(x_i^*, x_j^*) = k_{x_i^* x_j^*} - \mathbf{k}_{Xx_i^*}^\top (K_{XX} + \Sigma)^{-1} \mathbf{k}_{Xx_j^*},$$

$X = (x_1, \ldots, x_n)$ is the set of training points, $K_{XX}$ is the kernel matrix at $X$, and $\Sigma$ is the noise covariance.[2] LOVE achieves fast (co-)variances by decomposing $K_{XX} + \Sigma = RR^\top$ in $\mathcal{O}(r\nu(K_{XX}))$ time, where $R \in \mathbb{R}^{n \times r}$ and $\nu(K_{XX})$ is the time complexity of a matrix vector multiplication $K_{XX}v$. This allows us to compute the second term of the predictive covariance as:

$$k_{f|\mathcal{D}}(x_i^*, x_j^*) = k_{x_i^* x_j^*} - \mathbf{k}_{Xx_i^*}^\top R^{-\top} R^{-1} \mathbf{k}_{Xx_j^*},$$

where $R^{-1}$ denotes a pseudoinverse if $R$ is low-rank.[3] The main operation to perform is decomposing $\tilde{K}_{XX} = RR^\top$, where $\tilde{K}_{XX} := K_{XX} + \Sigma \in \mathbb{R}^{n \times n}$. Computing this decomposition can be done from scratch in $\mathcal{O}(nr^2)$ time. After forming $R$, additional $\mathcal{O}(nr^2)$ time is required to perform a QR decomposition of $R$ so that a linear least squares systems can be solved efficiently (i.e., approximate $R^{-1}$). $R$ and its QR decomposition are referred to as the LOVE *cache*.

Figure 2: CPU times for constructing fantasy model and evaluating its posterior at a single point (variance negligible relative to the mean).

**Algorithm 1:** Multi-Step Tree Evaluation

VALUE($\mathcal{M}_t$, $\boldsymbol{X}_t$, $\mathcal{D}_{t-1}$):
    $y^*_{t-1} = \max_{(x,y) \in \mathcal{D}_{t-1}} y$
    **return** $\mathbb{E}_{y \sim \mathcal{M}_t(\boldsymbol{X}_t)}\big[(y - y^*_{t-1})^+\big]$
STEP($\alpha_t$, $\mathcal{M}_t$, $\boldsymbol{X}_{t:k}$, $\boldsymbol{Z}_{t:k}$, $\mathcal{D}_{t-1}$):
    $\alpha_{t+1} = \alpha_t + $ VALUE($\mathcal{M}_t$, $\boldsymbol{X}_t$, $\mathcal{D}_{t-1}$)
    **if** $t = k - 1$ **then**
        **return** $\alpha_{t+1}$
    $Y_t = $ CORRELATE($\mathcal{M}_t(\boldsymbol{X}_t)$, $\boldsymbol{Z}_t$)
    $\mathcal{M}_{t+1} = $ FANTASIZE($\mathcal{M}_t$, $\boldsymbol{X}_t$, $\boldsymbol{Y}_t$)
    $\mathcal{D}_t = \mathcal{D}_{t-1} \cup \big\{(\boldsymbol{X}_t, \boldsymbol{Y}_t)\big\}$

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

[7]Making sure not to perform unnecessary copies of the initial data, but instead share memory when possible.

[8]We can also observe interesting behavior on the GPU, where inference for 64 training points is faster than for 32 (similarly, for 256 vs. 128). We believe this is due to these sizes working better with the batch dispatch algorithms on the PyTorch backend.

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

# Appendix to:

# Efficient Nonmyopic Bayesian Optimization via One-Shot Multi-Step Trees

## A Proof of Proposition 1

If we instantiate a different version of $x_t, t = 2, \ldots, k$ for each realization of $\mathcal{D}_{t-1}^{j_1 \ldots j_{t-1}}$, we can move the maximizations outside of the sums:

$$
\bar{v}_k(x \,|\, \mathcal{D}) = v_1(x \,|\, \mathcal{D}) + \max_{\mathbf{x}_2, \mathbf{x}_3, \ldots} \left\{ \frac{1}{m_1} \sum_{j_1=1}^{m_1} v_1(x_2^{j_1} \,|\, \mathcal{D}_1^{j_1}) + \frac{1}{m_1 m_2} \sum_{j_1=1}^{m_1} \sum_{j_2=1}^{m_2} v_1(x_3^{j_1 j_2} \,|\, \mathcal{D}_2^{j_1 j_2}) + \cdots \right\}. \tag{8}
$$

We make the following general observation. Let $G(x) := \max_{\tilde{x} \in \tilde{\Omega}} g(x, \tilde{x})$ for some $g : \Omega \times \tilde{\Omega} \to \mathbb{R}$. If $(x^*, \tilde{x}^*) \in \arg\max_{(x, \tilde{x}) \in \Omega \times \tilde{\Omega}} g(x, \tilde{x})$, then $x^* \in \arg\max_{x \in \Omega} G(x)$. The result follows directly by viewing $v_k(x \,|\, \mathcal{D})$ as $G$ and the objective on the right-hand-side of (6) as $g$.

## B One-Shot Optimization of Lower and Upper Bounds

As described in the main text, the non-adaptive approximation for pseudo multi-step lookahead corresponds to a one-shot optimization of a lower bound on Bellman's equation. Here we show this from a different perspective and also demonstrate that the multi-step path approach can be viewed as one-shot optimization of an upper bound depending on the implementation.

We can generate a lower-bound on the reparameterized (5) by moving maxes outside expectations:

$$
v_k(x \,|\, \mathcal{D}) \geq v_1(x \,|\, \mathcal{D}) + \mathbb{E}_z \left[ \max_{x_2, x_3, \ldots, x_k} \mathbb{E}_{z_2, z_3 \ldots z_{k-1}} \sum_{i=2}^{k} v_1(x_i \,|\, \mathcal{D}_{i-1}) \right] \tag{9}
$$

$$
= v_1(x \,|\, \mathcal{D}) + \mathbb{E}_z [\max_X V_1^{k-1}(X \,|\, \mathcal{D}_1)], \tag{10}
$$

where we recognize the second line as the objective for the non-adaptive approximation.

We similarly generate an upper-bound on the reparameterized (5) by moving maximizations inside the expectations as follows:

$$
v_k(x \,|\, \mathcal{D}) \leq v_1(x \,|\, \mathcal{D}) + \mathbb{E}_{z, z_2, z_3, \ldots, z_{k-1}} \left[ \max_{x_2, x_3, \ldots, x_k} \sum_{i=2}^{k} v_1(x_i \,|\, \mathcal{D}_{i-1}) \right]. \tag{11}
$$

This equation includes an expectation of paths of base samples $z, z_2, z_3, \ldots, z_{k-1}$ rooted at $x$. Consider one-shot optimization of the right-hand side of this equation with $m$ base sample paths:

$$
x^*, \mathbf{x}_2^*, \mathbf{x}_3^*, \ldots, \mathbf{x}_k^* = \arg\max_{x, \mathbf{x}_2, \mathbf{x}_3, \ldots, \mathbf{x}_k} \left\{ v_1(x \,|\, \mathcal{D}) + \frac{1}{m} \sum_{j_1=1}^{m} \sum_{i=2}^{k} v_1(x_i^{j_1} \,|\, \mathcal{D}_{i-1}^{j_1}) \right\}. \tag{12}
$$

Comparing to (6), the two expressions are identical when $m_1 = m$, and $m_2 = m_3 = m_4 \ldots = m_{k-1} = 1$. In the main text, our approach through Gauss-Hermite quadrature produces base samples $z_2 = z_3 = z_4 = \ldots z_{k-1} = 0$, which does not correspond to an approximation of (11). However,

when we use Monte-Carlo (or quasi-Monte Carlo) base samples for $z$, the one-shot optimization for multi-step paths does correspond to (11).

One can also derive analogous sample-specific bounds on one-shot trees defined by a given set of base samples. Forcing decision variables at the same level in the tree to be identical is the fixed sample equivalent of moving a max outside an expectation, and enforces a lower-bound on the specific one-shot tree. Moving a max inside the expectation also has a fixed sample equivalent: splitting the decision variable based on each base sample for that expectation, replicating the descendant tree including the base samples and decision variables, and then allowing all of these additional decision variables to be independently optimized produces an upper-bound on the tree.

## C   Implementation Details

**Optimization.** Because of its relatively high dimensionality, the multi-step lookahead acquisition function can be challenging to optimize. Our differentiable one-shot optimization strategy enables us to employ deterministic (quasi-) higher-order optimization algorithms, which achieve faster convergence rates compared to the commonly used stochastic first-order methods [1]. This is in contrast to zeroth-order optimization of most existing nonmyopic acquisition functions such as rollout and GLASSES. To avoid computing Hessians via auto-differentiation, we use L-BFGS-B, a quasi-second order method, in combination with a random restarting strategy to optimize (6).

**Warm-Start Initialization.** In our empirical investigation, we have found that careful initialization of the multi-step optimization is crucial. To this end, we developed an advanced warm-starting strategy inspired by homotopy methods that re-uses the solution from the previous iteration (see Appendix D). Using this strategy dramatically improves the BO performance relative to a naive optimization strategy that does not use previous solutions.

**Gauss-Hermite Quadrature.** Instead of performing MC integration, we can also use Gauss-Hermite (GH) quadrature rule to draw samples for approximating the expectations in each stage [18, 30]. In this case, when using a single sample as in the case of "multi-step path", the sample value is always the mean of the Gaussian distribution.

We implemented multi-step fast fantasies in `GPyTorch` [8], and the multi-step lookahead acquisition function in `BoTorch` [1]. Our code is included as part of this submission, and will be made public under an open-source license.

## D   Warm-Start Initialization Strategy for Multi-Step Trees

Warm-starting is an established method to accelerate optimization algorithms based on the solution or partial solution of a similar or related problem, specifically in case the problem structure remains fixed and only the parameters of the problem change. This is exactly the situation we find ourselves in when optimizing acquisition functions for Bayesian optimization more generally, and optimizing multi-step lookahead trees more specifically.

Since the multi-step tree represents a scenario tree, one intuitive way of warm-starting the optimization is to identify that branch originating at the root of the tree whose fantasy sample is closest to the value actually observed when evaluating the suggested candidate on the true function. This sub-tree is that hypothesized solution that is most closely in line with what actually happened. One can then use this sub-tree of the previous solution as a way of initializing the optimization.

Let $\boldsymbol{X}_{0:k}^* := \{\boldsymbol{x}_i^*\}_{i=0}^k$ be the solution tree of the random restart problem that resulted in the maximal acquisition value in the previous iteration. For our restart strategy, we add random perturbations to the different fantasy solutions, increasing the variance as we move down the layers in the optimization tree (depth component that captures increasing uncertainty the longer we look ahead), and increasing variance overall (breadth component that encourages diversity of the initial conditions to achieve coverage of the domain). Concretely, supposing w.l.o.g. that $\mathcal{X} = [0, 1]$, we generate $N$ initial conditions $\boldsymbol{X}_{0:k}^1, \ldots, \boldsymbol{X}_{0:k}^N$ as

$$\boldsymbol{x}_i^r = (1 - \gamma_r)\Big((1 - \eta_i)\boldsymbol{x}_i^* + \eta_i \beta_i^r\Big) + \gamma_r u_i^r, \tag{13}$$

with $\beta_i^r$ and $u_i^r$ of the same shape as $\boldsymbol{x}_i^*$, with individual elements drawn i.i.d. as $\beta_i^r \sim \text{Beta}(1,3)$ and $u_i^r \sim U[0,1]$ for all $r, i$, and $\gamma_1 < \ldots < \gamma_N$ and $\eta_0 < \ldots < \eta_k$ are hyperparameters (in practice, a linear spacing works well).

# E    Fast Multi-Step Fantasies

Our ability to solve true multi-step lookahead problems efficiently is made feasible by linear algebra insights and careful use of efficient batched computation on modern parallelizable hardware.

## E.1    Fast Cache Updates

If $R$ were a full Cholesky decomposition of $\tilde{K}_{XX}$, it could be updated in $\mathcal{O}(n^2)$ time. This is advantageous, because computing the Cholesky decomposition required $\mathcal{O}(n^3)$ time. However, for dense matrices, the LOVE cache requires only $\mathcal{O}(n^2 r)$ time to compute. Therefore, to use it for multi-step lookahead, we must demonstrate that it can be updated in $o(n^2)$ time.

Suppose we add $q$ rows and columns to $\tilde{K}_{XX}$ (e.g. by fantasizing at a set $\mathbf{x} \in \mathbb{R}^{q \times d}$ of candidate points) to get:

$$\left[ \begin{array}{cc} \tilde{K}_{XX} & U \\ U^\top & S \end{array} \right],$$

where $U \in \mathbb{R}^{n \times q}$ and $S \in \mathbb{R}^{q \times q}$. One approach to updating the decomposition with $q$ added rows and columns is to correspondingly add $q$ rows and columns to the update. By enforcing a lower triangular decomposition without loss of generality, this leads to the following block equation:

$$\left[ \begin{array}{cc} \tilde{K}_{XX} & U \\ U^\top & S \end{array} \right] \approx \left[ \begin{array}{cc} R & 0 \\ L_{12} & L_{22} \end{array} \right] \left[ \begin{array}{cc} R & 0 \\ L_{12} & L_{22} \end{array} \right]^\top$$

From this block equation, one can derive the following system of equations:

$$\tilde{K}_{XX} = RR^\top$$
$$U = RL_{12}^\top$$
$$S = L_{12}L_{12}^\top + L_{22}L_{22}^\top$$

To compute $L_{12}^\top$, one computes $R^{-1}U$. Since in the LOVE case $R$ is rectangular, this must instead be done by solving a least squares problem. To compute $L_{22}$, one forms $S - L_{12}L_{12}^\top$ and decompose it.

**Time Complexity.** In the special case of updating with a single point, $(q = 1)$, $U \in \mathbb{R}^n$ and $S \in \mathbb{R}$. Therefore, computing $L_{12}$ and $L_{22}$ requires the time of computing a single LOVE variance $(R^{-1}U)$ and then taking the square root of a scalar $(S - L_{12}L_{12}^\top)$. In the general case, with a cached pseudoinverse for $R$, the total time complexity of the update is dominated by the multiplication $R^{-1}U$ assuming $q$ is small relative to $n$, and this takes $\mathcal{O}(nrq)$ time. Note that this is a substantial improvement over the $\mathcal{O}(n^2 q)$ time that would be required by performing rank 1 Cholesky. If in each of $k$ steps of lookahead we are to condition on $m$ samples at $q$ locations, the total running time required for posterior updates is $\mathcal{O}(nrqm^{k-1})$.

**Updating the Inverse.** The discussion above illustrates how to update a cache $R$ to a cache that incorporates $q$ new data points. In addition, one would like to cheaply update a cache for $R^{-1}$ without having to QR decompose the full new cache. From inspecting a linear systems / least squares problems of the form

$$\left[ \begin{array}{cc} R & 0 \\ L_{12} & L_{22} \end{array} \right] \left[ \begin{array}{c} x \\ y \end{array} \right] = \left[ \begin{array}{c} b \\ c \end{array} \right] \tag{15}$$

one can find that $x = R^{-1}b$ and $y = L_{22}^{-1}(c - L_{12}R^{-1}b)$. Therefore, an update to the (pseudo-)inverse is given by:

$$\left[ \begin{array}{cc} R & 0 \\ L_{12} & L_{22} \end{array} \right]^{-1} = \left[ \begin{array}{cc} R^{-1} & 0 \\ -L_{22}^{-1}L_{12}R^{-1} & L_{22}^{-1} \end{array} \right]. \tag{16}$$

**Practical Considerations.** Suppose one wanted to compute cache updates at $J$ different sets of points $\mathbf{x}^{(1)}, \ldots \mathbf{x}^{(J)}$, each of size $q$. For efficiency, one can store a single copy of the pre-computed

cache $R^{-1}$ and just compute the update term $P_k := \begin{bmatrix} -L_{22,(j)}^{-1} L_{12,(j)} R^{-1} & L_{22,(j)}^{-1} \end{bmatrix}$ for each of the $\mathbf{x}^{(j)}$. To perform a solve with the full matrix above and an $n + q$ size vector $\mathbf{v}$, one can now compute $R^{-1}\mathbf{v}[:n]$, $P\mathbf{v}$, and concatenate the two. Therefore, an efficient implementation of this scheme is to compute a batch matrix $P \in \mathbb{R}^{J \times q \times (r+q)}$ containing $P_1, \ldots, P_K$ given a single copy of $R^{-1}$. This allows handling multiple input points by converting a single GPyTorch GP model to a batch-mode GP with a batched cache, all while still only storing a single copy of $R^{-1}$.

## E.2  Fast Fantasies

**Batched Models.** For our work, we extend the above cache-updating scheme from GPyTorch to the multi-step lookahead case, in which we need to fantasize from previously fantasized models. To this end, we employ GPyTorch models with multiple batch dimensions. These models are a natural way of representing multi-step fantasy models, in which case each batch dimension represents one level in the multi-step tree. Fantasizing from a model then just returns another model with an additional batch dimension, where each batch represents a fantasy model generated using one sample from the current model's posterior. Since this process can also be applied again to the resulting fantasy models, this approach makes it straightforward to implement multi-step fantasy models in a recursive fashion.

**Efficient Fantasizing.** Note that when fantasizing from a model (assuming no batch dimensions for notational simplicity), each fantasy sample $y_t^{j_i}$ is drawn at the same location $x_t$. Since the cache does not depend on the sample $y_t^{j_i}$, we need to compute the cache update only once. We still use a batch mode GP model to keep track of the fantasized values,[7] but use a single copy of the updated cache $\tilde{R}^{-1}$. We utilize PyTorch's tensor broadcasting semantics to automatically perform the appropriate batch operations, while reducing overall memory complexity.

**Memory Complexity.** For multi-step fantasies, using efficient fantasizing means that the cache can be built incrementally in a very memory- and time-efficient fashion. Suppose we have a (possibly approximate) root decomposition for $R^{-1}$ of rank $r \leq n$, i.e. $R \in \mathbb{R}^{n \times r}$. The naive approach to fantasizing is to compute an update by adding $q_t$ rows and columns in the $t$-th step for each fantasy branch. For a $k$-step look-ahead problem, this requires storing a total of

$$N_{\text{naive}} = nr + \sum_{t=0}^{k-1} \prod_{\tau=0}^{t} m_\tau \left( n + \sum_{\tau=0}^{t} q_\tau \right) \left( r + \sum_{\tau=0}^{t} q_\tau \right) \tag{17}$$

entries. For fast fantasies, we re-use the original $R^{-1}$ component for each update, and in each step broadcast the matrix across all fantasy points, since the posterior variance update is the same. This means storing a total of

$$N_{\text{FF}} = nr + \sum_{t=0}^{k-1} \prod_{\tau=0}^{t-1} m_\tau\, q_t \left( r + \sum_{\tau=0}^{t} q_\tau \right) \tag{18}$$

entries. If $q_t = q$ and $m_t = m$ for all $t$, then this simplifies to

$$N_{\text{naive}} = nr + \sum_{t=0}^{k-1} m^{t+1} (n + (t+1)q)(r + (t+1)q) \tag{19a}$$

$$N_{\text{FF}} = nr + \sum_{t=0}^{k-1} m^{t}\, q\, (r + (t+1)q) \tag{19b}$$

Assuming $r = n$, we have $N_{\text{naive}} = \mathcal{O}(m^k(n + kq)^2)$ and $N_{\text{FF}} = \mathcal{O}(m^{k-1}q(n + kq))$.

**Scalability.** Figure 7 compares the overall wall time (on a logarithmic scale) for constructing fantasy models and performing posterior inference, for both standard and fast fantasy implementations. On CPU fast fantasies are essentially always faster, while on the GPU for small models performing full inference is fast enough to outweigh the time required to perform the additional operations needed for performing fast fantasy updates.[8] For larger models we see significant speedups from using fast fantasy models on both CPU (up to 22x speedup) and GPU (up to 14x speedup).

Figure 7: Fast Fantasy wall time comparisons (log-scale). Wall time measures constructing the fantasy model and evaluating its posterior at a single point. Estimated over multiple runs, with variance negligible relative to the mean estimates. Results were obtained on an NVIDIA Tesla M40 GPU — we expect to see even more significant speedups on more modern hardware.

Table 4: Results on seven real hyperparameter tuning functions.

|  | EI | 6.EI.s | 2-path | 3-path | 4-path | 6-ENO |
|---|---|---|---|---|---|---|
| LogReg | 0.981 | *0.989* | *0.986* | *0.987* | *0.985* | **0.992** |
| SVM | *0.955* | 0.953 | **0.962** | *0.959* | *0.957* | *0.957* |
| LDA | *0.884* | **0.885** | *0.884* | *0.884* | *0.880* | *0.884* |
| Robot pushing 3d | *0.858* | **0.873** | *0.858* | *0.865* | 0.848 | 0.840 |
| NN Cancer | 0.480 | 0.638 | 0.568 | *0.652* | *0.669* | **0.683** |
| NN Boston | 0.457 | 0.461 | *0.475* | *0.495* | **0.496** | *0.485* |
| Robot pushing 4d | *0.408* | *0.406* | **0.419** | *0.413* | *0.402* | 0.382 |
| Average | 0.717 | *0.744* | 0.736 | **0.751** | *0.748* | *0.747* |
| Average (EI <0.8) | 0.448 | 0.501 | 0.487 | *0.520* | **0.523** | *0.517* |

# F   Results on Real Functions

We again use the same set of seven real functions as in [15]. They are SVM, LDA, logistic regression (LogReg) hyperparameter tuning first introduced in [24], neural network tuning on the Boston Housing and Breast Cancer datasets, and active learning of robot pushing first introduced in [27], and later also used in [19]. These functions are pre-evaluated on a dense grid. Log transform of certain dimensions of SVM, LDA, and LogReg are first performed if the original grid is on log scale. We follow [5, 6] and use a random forest (RF) surrogate model to fit the precomputed grid, and treat the predict function of the trained RF model as the target function. We find that the RandomForestRegressor with default parameters in scikit-learn can fit the data well, with cross validation $R^2$ mostly over 0.95. A Python notebook is included in our attached code reporting the RF fitting results.

Table 4 shows the results. The functions are arranged in decreasing order of EI GAP values. 6.EI.s is the best reported BINOCULARS variant in [15] for these functions. We only show results for $k$-path ($k = 2, 3, 4$) and 6-ENO. We can see when the function is "easy" (e.g., EI GAP > 0.8), there is almost no difference among all these methods. If we only average over the "hard" ones, we see a more consistent and significant pattern as shown in the last row of Table 4. We also plot the GAP curve vs. iterations for the three harder functions in Figure 8. Note the improvement of our method over baselines is statistically significant for NN Boston, despite the somewhat overlapping error bars in Figure 8(a). The improvement on NN Cancer is more evident.

Figure 8: GAP vs. #iterations on two neural network hyperparameter tuning functions. (a) regression network tuning on the Boston housing dataset. (b) classification network tuning on the breast cancer dataset.

Figure 9: GAP vs. iterations on individual synthetic functions.

# G Detailed Results on Synthetic Functions

In Figure 9, we show the GAP vs. iteration plot for each individual synthetic function. We can see our proposed nonmyopic methods outperform baselines by a large margin on most of the functions, especially on shekel5 and shekel7.