[Reviews · NeurIPS 2020]

Review 1

Summary and Contributions: A new approach is investigated to optimize multi-step lookahead Bayesian optimization acquisition functions, with a focus on the Expected Improvement. The key idea is to leverage the reparametrization trick to express a discretization of the multi-step EI as a function of all points coming into play in the next steps of the considered sequence, indexed by the total number of fantasized responses at those points. The big advantage of the reparametrization trick is that it enables explicitly disentangling the effect of those future observations from the candidate points, hence enabling to tackle the problem as a function of all candidate points only, given values of the so-called base samples. From there, automatic differentiation can be used to optimize over all points at stake, from which the optimal next point can be extracted, etc. Numerical experiments suggest that the proposed method(s) achieve(s) comparable or competitive results to state-of-the-art multi-step EI approaches.

Strengths: I am impressed by the key idea. It transforms an intractable problem into a higher-dimensional one that lends itself better to traditional optimization approaches. The results are very encouraging!

Weaknesses: The mathematical formalism is not always ideally treated, and in particular the "propositions" are quite handwavy. Besides this, the choice of base samples is not much discussed, while given the number of values strongly limited by the combinatorial explosion, one could expect that tuning well those values could be of substantial value here. Finally, I did not find Section 4 of utmost importance to the contribution, and I guess it could be sent to appendix so that more space could be dedicated to formalizing and explaining better the key ideas of the paper.

Correctness: Beyond minor issues mentioned next, I did not stumble across major issues (yet see comment above on mathematical formalism and, in particular, propositions)

Clarity: Yes, I found the paper well-written.

Relation to Prior Work: Yes, I found the paper well documented, and I got the impression that the work was fairly situtated with respect to previous contributions on the topic.

Reproducibility: Yes

Additional Feedback: Issue in eq. (4) with the domain of the second term in the RHS. Issue in eq. (5) in the index of the first expectation (y_1?) Notation discrepancy wrt pseudo-inverse in Section 4 ? ^{\dag} or ^{-}? Line 146, big oh of n^2? ############################# POST REBUTTAL ################### I originally gave the paper a score of 7 because I find the core idea brilliant and the proof of concept promising indeed. Of course, a number of practicalities need to be adressed to make of the proposed approach something broadly applicable. Now, I feel that the author's rebuttal is valuable and clarifies things regarding several concerns raised by the reviewing team; naturally not all of them with the same level of attention though. In particular, I was a bit disappointed that the authors did not clarify/investigate more how the way base samples are chosen/sampled could affect performances. Also, the rebuttal did not really convince me that the topics covered in section 4 do deserve dedicating that much room in the body of the article, while expanding other points and sections could in my opinion benefit the paper more. For these reasons, I do not feel like increasing my score. To me it is a good paper mostly because of the pioneering idea at its core. But it fails being a truly excellent paper because of not going deeper neither in formal/theoretical nor in advanced computational directions. Yet, as authors responded, some of these aspects could inspire entire follow-up works in their own right. Whether or not the paper gets accepted at this conference, and I am in any case looking forward to see some version of this work and offsprings thereof being published in the future.


Review 2

Summary and Contributions: The paper presents a new non-myopic algorithm for Bayesian optimization (BO). Non-myopic approaches to BO are a notoriously hard problem because the analytic expressions are intractable, and numerical evaluations suffer from infinite state-action spaces. Existing non-myopic algorithms are therefore either shortsighted or employ simplifying assumptions to increase the look-ahead depth. The latter is not ideal as increasing depth amplifies model errors. New algorithms and insights are therefore highly demanded by the community in the quest to improve existing BO approaches. Such insights may lead to large impact on many real-word problems that optimize noisy and expensive-to-evaluate functions. This paper provides a new numerical approach that tackles the intractable formulation head-on. Key novelty is a new formulation of the look-ahead utility function for jointly optimizing future sampled trajectories, and is evaluated in parallel on accelerated hardware. Since all decision variables are jointly optimized without nested maximization-optimization recursions, the tree formulation allows for gradient-based optimization. The paper also employs a numerically fast GP inference method based on the Lanczos Variance Estimate algorithm. With these insights, the authors are able to run the algorithm for horizons of up to four steps, albeit with a modest number of samples per step of 10 5, and 3, respectively. Benchmark results on selected synthetic functions are generally better than existing algorithms, while on real-word functions the results are on par with previous non-myopic implementations.

Strengths: - The main strength of the algorithm is that it can be efficiently implemented with automatic differentiation software and parallelized on accelerated hardware, thereby allowing to increase the look-ahead depth up to four steps. - The algorithm evaluates the utility function directly without making use of simplifying assumptions. This is both a strength and weakness. The strength relies on the unbiased evaluation of the utility function, while the lack of theoretical insight and inner workings of this black-box numerical approach capture its weakness. - The selected benchmark results are promising. For most synthetic and real benchmark functions, the algorithm generally performs on par or slightly better than previous non-myopic implementations. However, I find the selection of benchmarks rather restricted and unrepresentative. Details are given in the “weaknesses” section.

Weaknesses: - Lack of theoretical insight: despite the clever numerical implementation, the paper provides little theoretical insight into non-myopic BO. It is, in fact, not clear at all whether the crude numerical approximation of the utility function is sufficiently accurate to yield performance improvements over the *best* myopic algorithms [the benchmarks only provide comparisons with EI as noted below]. Similarly, there is little insight to parameter selection such as sample size. - Lack of performance guarantees: A large amount of work has gone into improving the numerical implementation, but no clear statement about performance guarantees is made. For instance, to what extent does an additional horizon depth guarantee performance improvement? Naively I would expect this guarantee to hold, but the “shekel7” benchmark function shows that, surprisingly, the 4-step version performs worse than the 3-step version. Another question to consider: what is the tradeoff between sample size and search depth? a

Correctness: As far as I can follow, the method and the claims related to the method are correct.

Clarity: The paper is overall well motivated and clearly written with only sections 4.1 and 4.2 being difficult to follow (see “Additional feedback”)

Relation to Prior Work: Relation to previous work is presented well.

Reproducibility: Yes

Additional Feedback: Update after rebuttal: After reading the other reviews and the authors' response, I changed my score to 6. It didn't fully realize the potential of the underlying idea (and its first successful implementation) and focussed to much on the empirical evaluation, which are not the main focus here. If the authors could highlight this, I think readers will have different expectations which are more aligned with the authors' intent. -------------------------------- - I believe the one-sided sign rank test to be the incorrect statistical test in Table 2. According to Demsar (2006, “Statistical Comparisons of Classifiersover Multiple Data Sets”) the Friedman test should be used, if the same random seed was used across the different methods, i.e. if the first run of EI started with the same 2d random points as the first run of ETS (and all the other methods). If that was not the case, I think the Kruskal-Wallis test would be appropriate here. - On line 165 it is stated that “In appendix E we show that speedups on a GPU are even higher”. I do not see such “higher” speedups in appendix E – I would appreciate, if the authors could elaborate on this. - I find sections 4.1 and particularly 4.2 hard to follow. It is not clear from the discussion what challenge the authors are trying to solve: which inference complexity can be achieved with existing state-of-the-art algorithms, and what is the contribution of this work. I find the discussion to be incoherent and fragmented. For instance, the authors argue they would like to improve on the Cholesky decomposition bottleneck O(n^3), and the LOVE cache allows a complexity reduction to O(r n^2). It is then stated that “Therefore, to use it usefully for multi-step lookahead, we must demonstrate that it can be updated in O(n^2) time.” – but it is not clear why. Finally they conclude that “A rank r + q decomposition R-1 of the inverse can be computed from R in O(nrq) time”, but no discussion around this result is provided.


Review 3

Summary and Contributions: Lookahead strategies have the dubious honor of being both a well-understood theoretical ideal and a practical nightmare. Whereas various recent works have focused on approximating this ideal, few have investigated methods for facilitating their use in the real world. This paper succeeds to the extent that it does so, but falters elsewhere. The authors propose two primary changes to subroutines common to many lookahead techniques, namely: (a) joint optimization of decision variables in a tree-structured integral approximator (b) exploitation of fast linear algebra. On both fronts, more detailed analysis would be greatly appreciated. Ultimately, the paper is held back by conflicting priorities. The authors' technical contributions advance solving (i.e. computing and optimizing) lookahead strategies. Some minor details aside, the proposed methods are generic (which is good). Much of the paper however (incl. the experiments) focuses on comparing a bevy of closely related acquisition functions. Such comparisons are problematic since they (i) are arguably orthogonal to the authors' main contributions (ii) lack the necessary depth to be convincing. If the authors delve deeper into the "theory for the practice" and communicate findings that benefit solving for a broad class of acquisition functions --- or, better yet, nested integrals defined w.r.t. GP posteriors --- I can see this becoming a truly excellent paper. For me to reverse my decision, the existing experiments need to be fleshed out and the following key points need to be addressed (via theoretical analysis and/or proper ablation): (1) Initialization strategies (i.e. warm-starting) (2) Joint vs. zeroth-order optimization (3) Treatment of branching factors m_{k} (4) Quadrature vs. vanilla Monte Carlo (esp. for "multi-path")

Strengths: Improving the practicality of solving for GP-based lookahead problems can have significant repercussions for BO (and other sequential decision-making scenarios). I think that this is a very good research direction and encourage the authors to pursue it further. So long as a good initialization strategy is available, joint optimization of decision variables seems very sensible and could become the method of choice in many cases.

Weaknesses: Jointly tuning all decision variables poses a very difficult optimization problem. The authors acknowledge this fact and clearly state that carefully initializing said variables is crucial. I would have expected this issue to be thoroughly studied in the body of the paper. Instead, a fairly intricate heuristic involving multiple hyperparameters is presented in appendices sans ablation. The experiments section seems rushed. Line 235 reads "Some entries are omitted from the table and plots for better presentation". Please report these results. As is, this omission is potentially misleading? For example, line 240 claims that "2-,3-,4-step outperform all baselines by a large margin" --- which is seemingly only true if we ignore 12-ENO? This issue recurs in appendices where multi-step methods are absent and 12- baselines become 6- ones? Overall, these omissions unfortunately casts doubt on the authors' findings. *Correction: I mistook ENO for ENS. As is, the argument about "hard" problems being better targets for nonmyopic acquisition functions seems rather hand wavy. If BO is to be used as a black-box optimizer, then it is necessary to measure its performance on different types of problems --- even those deemed unfavorable to nonmyopic strategies.

Correctness: ln 148: linear time -> quadratic time? ln 149: quadratic time -> O(rn^{2})? ln 240: "2-,3-,4-step outperform all baselines" -> "... most baselines"? Spectral Kernel Interpolation -> Structured Kernel Interpolation

Clarity: The paper is sufficiently well-written, but becomes disorienting at times. This issue is partially attributable to the authors' efforts to unify their approach with existing methods, which necessitates which necessitates introducing otherwise extraneous concepts and terms.

Relation to Prior Work: If I am not mistaken, the proposed multi-step acquisition function is effectively equivalent to that of [1] when v_{1}(...) = EI(...), differing only in terms of an additive constant. This connection is alluded to but not explicitly stated. [1] Gaussian Processes for Global Optimization, Osborne et al (2009)

Reproducibility: Yes

Additional Feedback: It is mildly unfortunate that the authors so heavily emphasize Bo/GPyTorch and Lanczos methods. Balancing this focus with attention to common alternatives (e.g. use of Cholesky factors) would help the paper to reach a broader audience. I am curious about the numerical stability for explicitly computing (and updating) the pseudoinverse R^{-1}. Have you considered updating the (thin) QR decomposition of R instead (see, e.g., [2])? Have you considered using a zeroth-order method to initialize the decision tree (or a suitable proxy) and, subsequently, use gradients to jointly fine-tune? Have you considered updating the prior mean at each node in the decision tree? Assuming use of a constant mean, the MLE update to the prior mean is simply a cumulative moving average. [2] Methods for Modifying Matrix Factorizations, Gill et al (1974)

[Author Response · NeurIPS 2020]

We thank the reviewers for their insightful comments. First, we want to highlight that the main contribution of this paper is a novel and efficient realization of general multi-step lookahead BO, which has *never been successfully attempted* before. This is a long-standing and notoriously hard problem (see, e.g., [22]) and we now filled in the gap. The generality of our approach with differentiable, multi-step trees allows swapping in different utility functions to get different nonmyopic BO policies, or to solve an entirely different problem such as Bayesian quadrature. We will first address some common issues, then individual ones.

**Number of samples** (branching factor $m_i$): Indeed this part could benefit from further analysis. From our results, we see even one sample (i.e., multi-path) produced reasonable results. We attempted to systematically study how $m_i$ would change the performance, only to realize it was much more complicated than we thought. First, more samples means higher approximation accuracy, however, it also increases the dimension of the optimization dramatically, which makes it harder to optimize; poorer optimization could counter the benefit of higher accuracy. Second, even if we could optimize it perfectly, would it necessarily be better to have a more accurate approximation? In theory, yes, but in practice, given that "all models are wrong" and hence more lookahead may hurt (see [33]), it is not necessary that more accurate approximation of a "wrong" expected utility will lead to better results – perhaps a rough estimate is all we need in practice. That said, we are not arguing against lookahead policies, but merely pointing out that some seemingly simple questions may not be so simple. In-depth analysis could merit a standalone paper.

**Sec. 4**: We realize that adding this dense section makes the paper more challenging to read. However, it presents essential techniques that make our method efficient. We will improve the clarity of the presentation.

**R1:** Proposition 1 is simply saying $\max_{x,y} f(x,y) = \max_x \max_y f(x,y)$, which is a key idea for one-shot optimization. We assume the decision variable of the current step to be $x = x_1, y = y_1$.

**R3: Theoretical guarantees**: In expectation, full-lookahead (i.e., the optimal policy) is guaranteed to be better than greedy by definition. In general, it can be very challenging to prove or disprove $k$-step rolling horizon is always better than 1-step in expectation. Note all recent contributions to nonmyopic BO are also lacking in this regard [22,11,18,15].
**Higher speedups on GPU**: this is inaccurate, we intended to say "speed" is higher.

**R5**: **Warm-start**: As we noted, the joint objective is a highly complicated high-dimensional function (up to around one thousand dimensions), and how to optimize it is critical. We use a perturbed version of the solution from the previous iteration to initialize the optimization, inspired by the conjecture that the surface would not change too much and there should be a nearby optimum. We did experiment without warm-start, for example, the average GAP of 3-step on the synthetic functions is about

|  | EI | 2-path | 3-path |
|---|---|---|---|
| branin | *0.993* | **0.997** | *0.989* |
| rosenbrock2 | *0.972* | **0.984** | *0.968* |
| hartmann3 | **1.000** | 1.000 | 1.000 |
| hartmann6 | 0.974 | **0.976** | *0.975* |
| levy2 | **0.987** | *0.982* | *0.970* |

0.63, which is much worse than with warm-start (0.747, see Table 2 of the paper) (note that this is still much better than EI). We found warm-starting to result in very large improvements. Further study is needed to fully understand its effectiveness. **Gaussian-Hermite (GH) vs. MC quadrature**: We used GH in our experiments only to follow previous work on nonmyopic BO (see e.g., [18, 32]), and it's also a well-established quadrature rule for 1d integral against a Gaussian. We did experiment with quasi-MC (Sobol) for multi-step and multi-path, for multi-step GH tends to be better, but for multi-path we did not observe considerable differences. We will present both results in the camera-ready version. **Relation to prior work**: The derivation of the optimal policy is simply an instantiation of the well-known Bellman equation, and it has been derived in several previous papers in various formats, including [18, 22, 15]. We are not inventing multi-step lookahead, but providing a novel and efficient solution technique with promising empirical results. **Omitted entries**: Fig. 4(a) plots the last two rows of Table 2, and the omitted entries are actually complementary. We arranged it this way only to avoid oversized table and cluttered plot. We can add everything back to reduce confusion. **Outperform all baselines**: Here baseline means EI, ETS and 12.EI.s; 12-ENO is a variant of our method. We will clearly delineate our methods vs baselines in the camera-ready. **Selection of benchmarks**: We fully adopted the benchmarks used in [15], and we also argue that nonmyopic methods have significant advantage over myopic methods on "hard" function, but not worse on "easy" ones. To address your concern, we arbitrarily chose some commonly used "easy" benchmarks and ran EI, 2-path, 3-path for 100 repeats. Results are shown in the above table. We can see the tested multi-step variants are never significantly worse than EI. We are happy to add the results of all 31 benchmarks in [15] to the camera-ready version. **Zeroth-order initialization**: Thanks for your interesting suggestion, we can certainly try using DIRECT, CMA-ES or even BO to initialize the optimization. **Emphasis on Bo/GPyTorch**: We stress that one-shot optimization is not tied to any underlying implementation framework. However, one of our contributions is a generic implementation that was facilitated to a large degree by the abstractions, efficient inference methods, and auto-differentiation capabilities in Bo/GPyTorch.

Again, we appreciate all the concerns raised by the reviewers and will certainly add relevant details to the camera-ready version to address them. However, most of the main points are actually deep questions that deserve future research. Our contributions are fundamental and provide a versatile framework that enables studying these important questions.

[Meta-Review · NeurIPS 2020]

The paper has been actively discussed by the reviewers and the AC (who also carefully read the submission). The rebuttal was also fully exploited. As a summary (with pros+ and cons-): + Relevant problem + Well written and clearly presented + Novel to the best of our knowledge + Original idea that is likely to inform future research + Non-trivial linear algebra result (Prop. 2) whose resulting fast computation is properly illustrated + Favorable comparison with existing approaches (again, with the results well communicated) + Availability of the code (the emphasis on Bo/GPyTorch can toned down as explained in the rebuttal, e.g., "since low-level abstractions/primitives were readily available in Bo/GPyTorch, we have focused on this particular package...") - Insufficient empirical study of the warm-starting strategies (the rebuttal has helped a bit in this respect) (A) - Insufficient empirical study of the choice of the samples/branching factors (the rebuttal was disappointing here, eluding a bit the core questions) (B) - Insufficient empirical study of the impact of the optimizer used (e.g., also zero-th order optimizers) (C) - Missing empirical discussions about Gaussian-Hermite vs. MC quadrature: as discussed in the rebuttal, a comparison should be included in the final version (D) - Discussion of the experimental results not enough in depth, e.g., "It is not clear at this point whether this is because we are not optimizing the increasingly complex multi-step objective well enough or if additional lookahead means increasing reliance on the model being accurate, which is often not the case in practice [33]." (E) --> the authors should consider a case where the model is correct (e.g., draws from a GP) and properly analyze the phenomenon All in all, the paper is recommended for acceptance. However, I urge the authors to take concrete actions to develop more in-depth studies/ablations for the points A-B-C and E by the time of the final version of the paper. Moreover, the problems (beyond the analysis expected for A-B-C and E) that are left open should be better communicated in the body of the text.